# Relationships between Iraqi Rice Varieties at the Nuclear and Plastid Genome Levels

**DOI:** 10.3390/plants8110481

**Published:** 2019-11-07

**Authors:** Hayba Badro, Agnelo Furtado, Robert Henry

**Affiliations:** Queensland Alliance for Agriculture and Food Innovation, University of Queensland, Brisbane, QLD 4072, Australia; haybaq@yahoo.com (H.B.); a.furtado@uq.edu.au (A.F.)

**Keywords:** rice (*Oryza sativa*), evolutionary relationships, chloroplast genome, nuclear genome, phylogeny

## Abstract

Due to the importance of the rice crop in Iraq, this study was conducted to determine the origin of the major varieties and understand the evolutionary relationships between Iraqi rice varieties and other Asian rice accessions that could be significant in the improvement of this crop. Five varieties of *Oryza sativa* were obtained from Baghdad/Iraq, and the whole genomic DNA was sequenced, among these varieties, Amber33, Furat, Yasmin, Buhooth1 and Amber al-Baraka. Raw sequence reads of 33 domesticated Asian rice accessions were obtained from the Sequence Read Archive (SRA-NCBI). The sequence of the whole chloroplast-genome was assembled while only the sequence of 916 concatenated nuclear-genes was assembled. The phylogenetic analysis of both chloroplast and nuclear genomes showed that two main clusters, Indica and Japonica, and further five sub-clusters based upon their ecotype, *indica*, *aus*, *tropical*-*japonica*, *temperate*-*japonica* and *basmati* were created; moreover, Amber33, Furat, Yasmin and Buhooth1 belonged to the *basmati*, *indica* and *japonica* ecotypes, respectively, where Amber33 was placed in the *basmati* group as a sister of cultivars from Pakistan and India. This confirms the traditional story that Amber was transferred by a group of people who had migrated from India and settled in southern Iraq a long time ago.

## 1. Introduction

Rice is grown in a wide range of environments worldwide, however, most of the world’s rice is cultivated and consumed in Asia [1,2,3]. Iraq has favorable agricultural conditions for rice cultivation, where rice is a staple food for the majority of the Iraqi people [4]. In Iraq, rice grows as a summer crop, and there are a number of traditional, introduced and improved rice varieties that are cultivated in the central and southern region, as well as in the valleys of northern Iraq [1].

The variety Amber is the most important local Iraqi rice variety and is characterised by high quality in terms of taste (aromatic character) [1]. It has been cultivated in central and southern Iraq, especially in the marshes, for a long time. Anecdotal evidence suggests that Amber was introduced to the marshlands of southern Iraq when water buffalo breeding was introduced to the region by a foreign group from the south Asia, probably from the Indian subcontinent. This popular view was reported in a study by Al-Zahery et al. [5], that highlighted the paternal and maternal origin of the human population in the marsh areas, and observed marginal influences of Indian origin on the gene pool of an autochthonous population of the region. A number of rice varieties have also been introduced to Iraq since the middle of the last century to improve rice productivity [6]. IR8 was the first variety introduced in 1968 by the International Rice Research Institute (IRRI) (Philippines), it has high yield potential but the grain quality has not been high compared to Amber. Since aroma is one of the key traits in determining grain quality in rice [7], Amber became a control variety in the central and south regions of Iraq to assess the grain quality of introduced varieties [1]. Accordingly, Furat and Yasmin were also introduced from Vietnam to Iraq in the late 20th century because they are aromatic, tolerant to limited water, highly productive, and have high grain quality [4]. An understanding of the origin of local Iraqi rice and the genetic relationships between Iraqi rice and Asian domesticated rice will effectively guide Iraqi rice breeding (the aim of the current study). However, few studies have investigated Iraqi rice in general, and the origin and the evolution of Iraqi varieties, especially Amber, in particular [4,8,9].

Each living organism is the consequence of an evolutionary process [10]; therefore, it is imperative to enrich our perception of the evolutionary history of organisms and the relationships among them to guide their genetic improvement. Methods of determining evolutionary history (Phylogeny) have undergone many stages of development. Morphological markers maybe influenced by environmental factors and growth practices. More recent methods have used molecular markers which are independent of environmental factors [11], including techniques such as RFLP, AFLP, RAPD, SSR and ISSR along with morphological markers, to study phylogenetic relationships [12]. The development of high-throughput sequencing technology has revolutionised the study of genetics and evolutionary relationships. Most recently, through next-generation sequencing (NGS), whole-genome sequencing and re-sequencing have become available, so the investigation of the entire genome, rather than targeting precise regions, is now a real opportunity [13,14,15].

Every plant cell has three genomes—nuclear, chloroplast, and mitochondrial—that may differ in evolutionary history. The chloroplast genome is a maternal genome which is highly-conserved and not involved in recombination, therefore, it is the most commonly used tool to determine the origin and the evolutionary relationships among plant species [16,17,18,19]. However, sometimes, evolutionary analysis based on the chloroplast genome must be supported by nuclear genome-based analysis to achieve the most reliable results because the chloroplast genome can only represent the maternal evolutionary history with a slow evolutionary rate [20,21]. Phylogenetic analysis using the nuclear genome can deliver inconsistent trees due to recombination that may confuse phylogenetic resolution. However, this analysis provides greater insights into evolutionary relationships. Several studies have strongly suggested applying this analysis along with chloroplast phylogenetic analysis [18,19,22]. Many studies have applied phylogenetic analysis at both genome levels [23,24,25], and the results of most of these studies showed that the nuclear genome followed a different evolutionary history pattern to that of the chloroplast genome.

We reported the whole chloroplast genome sequences for Iraqi rice and compared them with the whole chloroplast sequences of other domesticated Asian rice varieties. This provided an important tool for estimating genetic distance and determining evolutionary relationships between rice accessions; the nuclear genomes also provided further information on the relationships between the varieties studied. The study aimed to determine the origin and evolution of Iraqi rice, especially Amber33.

## 2. Results

### 2.1. DNA Sequencing and Data Processing

The sequencing process of the five Iraqi varieties (Table 1) generated about 51 Gb of data containing 337 million of 151-bp paired-end reads. The minimum and the maximum number of reads were about 58 and 93 million reads with sequence depth ranging between 23× and 38× for Buhooth1 and Furat, in turn. When raw data was trimmed at the quality limit of 0.01, an average of 15% of the reads’ length and 9% of the number of reads were removed, thus the number of reads and data coverage reduced to the range between 53 and 86 million, and 18× and 30×, respectively (Appendix A). In terms of downloaded data (Table 2), the average length of raw reads was 83-bp, and the minimum and the maximum number of reads ranged between 43 and 117 million reads while the sequence coverage fluctuated between 10× and 26×. Finally, the number of reads and the data coverage of each of the data sets were assessed after trimming the raw reads at the quality limit of 0.01 (Appendix A).

### 2.2. Chloroplast Genome Assembly

Mapping all varieties against the reference, *O. sativa* sub sp. japonica *Nipponbare* “GenBank: GU592207.1”, under three various fraction settings clarified the most accurate and reliable mapping setting. The number of mismatches and gaps of each variety was virtually stable in all different settings (Appendix A). Indeed, this stability confirms that most of these variations were produced from actual differences between the sequences of samples and reference, not due to using different settings; based on that, setting number two (length fraction (LF) of 1 and similarity fraction (SF) of 0.8) was applied to the other steps of assembly, Improvement process (Imp). Moreover, three different settings of Word “W” and Bubble “B” size in *de novo* assembly generated a satisfactory number of contigs that cover the whole chloroplast genome area, around five large chloroplast-contigs with a length of more than 12 kb produced from each setting. Subsequently, four main regions of the chloroplasts, large single copy (LSC), inverted repeat A (IR A), small single copy (SSC) and inverted repeat B (IR B), were assembled successfully for all 38 varieties through a *de novo* assembly pipeline. The lengths of these regions were about 80 kb for LSC, 12 kb for SSC and 20 kb for IR A and IR B (Appendix A shows only Iraqi rice varieties). In manual-curation, the comparison between both sub-approaches of the chloroplast genome assembly pipeline showed no significant differences in terms of the number of variations; however, any minor conflicts were resolved by reference to the reads (Appendix A shows only Iraqi rice varieties). The minimum and maximum lengths of the whole chloroplast for all Iraqi varieties and downloaded accessions were 134,259 and 134,556 bp, respectively, while the coverages ranged from 839× to up to 11,466×, and the average coverage was 3818× (Table 3).

### 2.3. Phylogenetic Analysis of the Chloroplast Genome

Two phylogenetic approaches were used to analyse the multiple alignments of thirty-nine chloroplast genomes which had a total length of 134,535 bp. Although the result of both phylogenetic methods showed some minor alterations at the end of some subclades, the content of the main clades and subclades, which followed their ecotype classifications, were identical (Figure 1). Phylogenetic analysis of the chloroplast genome divided the thirty-nine rice accessions into two main clades, an Indica clade and a Japonica clade. The Indica clade (In) included most individuals under *indica* (6 accessions) and *aus* (5 accessions) ecotypes except two individuals, B009 and IRIS_313-10718. The Japonica clade contained two subclades, a main Japonica clade and a Basmati clade; the first subclade which was the main Japonica clade (Jap) included all *japonica* individuals (13 accessions) from the two subpopulations of *japonica* ecotype, *tropical* and *temperate*, while the second subclade, the Basmati clade (Bas), involved all individuals of *basmati* ecotype (6 accessions) and the excluded individuals from the first clade (Indica). Additionally, the Iraqi varieties were distributed as following: Furat, Yasmin and Amber al-Baraka into the Indica clade whereas Amber33 and Buhooth1 into the Japonica clade. Buhooth1 was close to accessions from *tropical japonica* ecotype more than accessions under *temperate japonica* ecotype, and interestingly, Amber33 was located within the Basmati subclade.

The multiple alignments of chloroplast genomes comprised 134,535 bp, the number of identical sites was 134,270 characters (99.8%) while the number of variable bases among all the accessions totaled 265 (0.2%). These 265 variable bases were sorted into 85 variation positions which were in turn grouped into four types of polymorphisms including single nucleotide polymorphism (SNP), multi nucleotide polymorphism (MNP), insertions (Ins) and deletions (Del) (Table 4). The most abundant polymorphism types among all accessions were SNPs. Out of 85 polymorphisms, 83%, 12% and 5% were located in the four main regions of the chloroplast genome, LSC, SSC and IR A and B, respectively (Appendix A).

Considering the variations identified, all thirty-nine rice accessions were sorted into three main groups: (1) Indica, (2) Japonica and (3) Basmati. As expected, the highest number of polymorphisms among the species studied (255 bases in 76 variant positions) was found in the Indica group, 11 accessions and 3 Iraqi varieties; within 76 variants, there was only one variation (1-bp deletion at position of 75990 bp) between *indica* and *aus* accessions. While the second largest number of variations (55 bases within 21 variant positions) was within the Basmati group, 8 accessions and one Iraqi variety. Part of the Basmati group, 4 accessions, showed unique polymorphisms (2 variable bases (SNPs) within 2 variant positions), three accessions were from Pakistan IRIS_313–8656, IRIS_313–11026, and IRIS_313–11021) and one from Iran (CX104). As expected, the Japonica group, 13 accessions along with the reference (*O. sativa* sub sp. japonica Nipponbare “GenBank: GU592207.1”) and one Iraqi variety, possessed the lowest number of polymorphisms (13 bases within 10 variant positions) (Appendix A). Most of the polymorphisms in the Japonica group belonged to only four accessions from *tropical japonica* (TrpJ) subpopulation, including CX352, IRIS_313–10073, CX243, and IRIS_313–11248.

Furthermore, a heat-map was drawn according to the number of variable bases (Appendix A); in this map, the two main clusters, Indica and Japonica, were clearly distinguished, whereas the Basmati group was comprised within the Japonica group. Within the Japonica group two individuals, 24:IRIS_313–11479 and 27:IRIS_313–11248, clearly showed the greatest distances among the rice accessions. This cluster surprisingly also included two individuals, 33:IRIS_313–10718 and 34:B009, from the *aus* and *indica* ecotype, respectively. There were no variable bases between a number of pairs (dark red in Appendix A) such as (3:IRIS_313–11152 and 4:IRIS_313–9505), (9:CX126 and 11:CX37), (9:CX126 and 13:CX25), (17:IRIS_313–10073, and 18:CX243), (19:Ref-GU592207.1 and 20:IRIS_313–11153), (20:IRIS_313–11153 and 21:IRIS_313–10373); and (30:IRIS_313–8656, 31:IRIS_313–11026 and 37:CX104); whereas the highest number of variable bases, 260 bases, was found between (14:CX227 and 24:IRIS_313–11479) (dark green in Appendix A). The smallest number of variable bases between Iraqi varieties and other domesticated rice accessions were 1, 3, 1, 6 and 4 bases, those bases were between Iraqi varieties: Amber33, Furat, Yasmin, Buhooth1, and Amber al-Baraka, and the following accessions: 28:IRIS_313-10670, 12:CX10, 3:IRIS_313–11152, 20:IRIS_313–11153 and 5:IRIS_313–10549, respectively (Appendix A).

### 2.4. Phylogenetic Analysis of the Nuclear Genome

Within a group of thirty-nine rice accessions, the multiple alignment of 916 concatenated nuclear genes was 621,012 bp in length; the minimum and maximum lengths were 616,099 and 616,393 bp, respectively (Table 3). The nuclear phylogenies using two different methods showed that the two main clusters, Indica and Japonica, and further five sub-clusters were based upon their ecotype, *indica*, *aus*, *tropical japonica*, *temperate japonica* and *basmati* (Figure 2). Unlike the results of the chloroplast phylogeny, the accessions of *indica*, and *aus* ecotypes were represented by two well-resolved subclades within the Indica clade. The Iraqi varieties, Furat and Yasmin, were found in the *indica* subclade while the rest of the Iraqi collection was grouped in the Japonica clade, where Amber33 acted as a sister to all the *basmati* varieties within the *basmati* subcluster, which included all accessions with the *basmati* ecotype. Buhooth1 was part of the *temperate japonica* subcluster that comprised accessions from the *temperate japonica* ecotype and the reference, *O.s japonica* cv. Nipponbare. Amber al-Baraka was a sister to both the Indica and Japonica clades; however, Geneious Tree Builder showed that it was close to the Indica clade, while MrBayes suggested that Amber al-Baraka was closer to the Japonica clade.

## 3. Discussion

Rice phylogeny has been extensively studied as a better understanding of the evolutionary relationships among rice species is critical for rice breeding programmes as well as comparative genomics studies. Recent advances in next-generation DNA sequencing (NGS) have improved the phylogenetic reconstruction of any plant species including *Oryza*. In this study, both plastid and nuclear genomes were assembled using NGS reads (whole genome DNA sequencing) to identify the phylogenetic relationships among Iraqi rice varieties and other accessions. According to Sims et al. [27], the accuracy of a genome assembly using NGS reads depends on many factors including sequencing depth (coverage) and the accuracy of the assembly pipeline. Therefore, even after trimming, the sequence coverage of the sequenced and downloaded accessions (Appendix A) was enough to ensure coverage of all the chloroplast and most of the nuclear genome, thereby guaranteeing a high-quality assembly.

A dual pipeline was applied to the assembly of the chloroplast genome in this study; this pipeline consisted of two procedures, mapping assembly (MA), and *de novo* assembly (*d*A). The comparison between the sequence of *de novo* and mapping consensus showed no significant differences in terms of the number of variations. Interestingly, the variety Yasmin showed no difference in both approaches with regard to a number of variations (Appendix A), but the length of the consensuses was different; this observation indicates that even when the number of copies of an insertion or deletion was similar, the number of bases that were inserted or deleted was diverse. Therefore, in agreement with an earlier study [28], a manual-curation step was critical in resolving any conflicts by reference to the reads. A pipeline of nuclear genes assembly was also developed in this study. This pipeline involved multiple tools on the CLC Genomics Workbench, unlike a previous study [25] that used different software packages to assemble the nuclear genes for phylogenetic analysis at the nuclear genome level. The number of genes selected to represent the nuclear genome in the phylogenetic analysis was only 916 genes with a length of 621,012 bp, considerably lower than that reported previously [25].

Phylogenetic analysis of the chloroplast genome sorted the thirty-nine rice accessions into two main clades, an Indica clade and a Japonica clade (Figure 1). The Indica clade (In) included most individuals under the *indica* and the *aus* ecotypes except for two accessions. Accessions from *indica* and *aus* ecotypes were not clearly distinct but were placed together in one clade; this was confirmed by the results of genetic polymorphism analysis that showed only one variation (1-bp deletion at the position of 75,990 bp; Appendix A) between the *indica* and *aus* accessions. The Japonica clade (Jap) contained two subclades, the main Japonica clade (Jap) which included all individuals from the two *japonica* subpopulations, and the Basmati clade (Bas) that included all basmati accessions as well as the individuals excluded from the first clade (Indica). Moreover, the presence of accessions from *aus* ecotype in the Indica clade as well as in the Basmati subclade within Japonica clade agrees with earlier outcomes [18] which indicated that the two different ecotypes, *indica* and *japonica*, might be involved in the origins of the maternal genome in two Korean *aus* landrace rices. This also agrees with the conclusion made by Civáň et al. [29]. which suggested that *aromatic* rice resulted from a hybridization between *japonica* and *aus*. Analysis of genetic polymorphisms at the chloroplast genome level revealed that the most abundant variation types were SNPs, 57% of 85 variants (Table 4). This analysis also showed 255 nucleotide differences within 76 variant positions between the *O.sativa* spp. *indica* and the *japonica* reference (GU592207) in agreement with the previous studies of Brozynska et al. [22] and Wambugu et al. [28].

At the nuclear genome level, the phylogenetic analysis using two different approaches sorted accessions from *indica* and *aus* ecotypes into two completely independent subclades within the Indica clade, unlike the result of the chloroplast phylogeny, whilst the second clade was a Japonica clade which included three sub-clusters *tropical japonica*, *temperate japonica*, and *basmati* (Figure 2). Accordingly, the findings of the evolutionary relationship based on nuclear and chloroplast data in the current study aligned with an earlier study by Garris et al. [30] which reported that the closest evolutionary relationships were between *indica* and *aus* groups, and among the *tropical japonica*, *temperate japonica*, and *aromatic* groups. In general, in the present study, the phylogenetic analysis at both genome levels, chloroplast and nuclear, showed relatively comparable evolutionary history patterns with insignificant differences at the end of clades, unlike other studies that recorded significant differences in evolutionary history pattern using both chloroplast and nuclear genomes (regardless of plant materials) [23,24,25,29]. Furthermore, the phylogenetic trees of both genomes, chloroplast and nuclear, constructed using different methodologies, were highly compatible. However, Amber al-Baraka showed slightly different relationships at the level of the nuclear genome according to the method used; where Geneious Tree Builder software placed Amber al-Baraka close to the Indica clade whereas Amber al-Baraka was closer to the Japonica clade and distant from the Indica clade by MrBayes software. This was unexpected and requires further investigation.

The phylogenetic analysis of both the chloroplast and nuclear genomes indicated that Amber33, Furat and Yasmin, and Buhooth1 belonged to *basmati*, *indica* and *japonica* ecotypes, respectively. Our results supported that Buhooth1 is an improved cultivar, where the nuclear phylogenies showed a divergent relationship to those deduced from the chloroplast genomes, analogous to *temperate japonica* subpopulation and *tropical japonica* subpopulation, respectively. Furat and Yasmin were introduced to Iraq from Vietnam [4], this was obvious by the results of phylogenetic analysis of the nuclear genome, but their chloroplast genome was closely related to accessions from China, India and Philippines. This may be explained by the breeding history of the genotype.

In this study, Amber33, which is local Iraqi variety, was placed in the *basmati* ecotype group as a sister of cultivars from Pakistan and India by analysing the evolutionary relationship at both levels of the genome. Based on distance analysis, the number of differences in the chloroplast genome between Amber33 and all accessions within the Basmati subclade was in the following order: 28:IRIS_313–10670 O (1 bp), 35:CX59 (1 bp), 30:IRIS_313–8656 (3 bp), 31:IRIS_313–11026 (3 bp), 37:CX104 (3 bp), 33:IRIS_313–10718 (3 bp), 32:IRIS_313–11021 (5 bp), 34:B009 (6 bp) (Appendix A); it can be accordingly concluded that Amber33 is closely related to accession from India which is visibly reflected in the observed phylogenetic tree (Figure 1). This confirms the popular tradition that says that the Amber variety was transferred by a group of people who had migrated from India (the Southeast) and settled in southern Iraq a long time ago.

Recently, the term Basmati has been used to indicate a long-grain and high-quality rice, but this name originally refers to *aromatic* rice because it was derived from the Sanskrit words “Vas” and “Matup” which stand for “aroma” and “ingrained from the beginning”, respectively, and then both words were combined making ‘Vasmati’ which changed to become ‘Basmati’ later on [31,32]. Therefore, the presence of Amber33 within the Basmati subcluster does not necessarily mean that it is a long grain cultivar; indeed, it is an aromatic medium-grain cultivar. Furthermore, Basmati is a group that can be described basically as the fifth isozyme group identified by Glaszmann [33], and it is closer to the *japonica* group than the *indica* [7,30,34]; this group is also phenotypically diverse as it includes both long or medium grain, and aromatic or nonaromatic varieties [7]. In many studies, this group is also known as the “*aromatic*” subpopulation [7,33], but most of the time it is known as “Group V” to avoid confusion. In this study, we refer to this group as “Basmati” according to Wang et al.’s study [26] which is the information resource of the downloaded accessions.

## 4. Conclusions

In the present study, we have assembled the whole chloroplast genome and the nuclear genome of the five Iraqi rice varieties, together with thirty-three domesticated Asian rice, to find the origin of Iraqi varieties, especially Amber33, and to gain insight into the evolutionary relations between Iraqi and domesticated Asian rice varieties. Our results suggest that the possibility of an Indian and/or Pakistani origin for Amber33; to evaluate this hypothesis, further historical biogeographical analyses are required. Moreover, further study on the chloroplast and nuclear genome in Iraqi rice varieties are required to determine the functional genome annotations that might be useful for future rice breeding programmes in Iraq.

## 5. Materials and Methods

### 5.1. Plant Materials

A total of five varieties of *Oryza sativa* were provided and tested by the Office of Agricultural Research, and Directorate of Seed Testing and Certification, Ministry of Agriculture, Baghdad, IRAQ, respectively. Among these varieties, one variety, Amber33, is local and one of the most highly valued varieties in Iraq because of its fragrance, and two varieties, Furat and Yasmin, were introduced from Vietnam; however, they are successfully cultivated in the central and southern regions of Iraq; while the other two, Buhooth1 and Amber al-Baraka, are improved varieties [4]. The plant materials used in this chapter are described in detail in Table 1.

### 5.2. Seed Germination and Growth

About 15 seeds of each individual, a total of 75 seeds, were first dehusked, and then placed in a container with plenty of liquid fertilizer, Flowfeed EX7, that was diluted to half concentration (full concentration is 0.5 g/1L) to break the dormancy phase; this method was the non-heat treatment method. Once the radicle emerged, the germinated seeds were transferred to a petri dish covered with a layer of tissue that was saturated with liquid fertilizer, and planted within three days. All the germination and planting processes were carried out under extremely restricted quarantine conditions in quarantine facilities.

### 5.3. DNA Extraction and Sequencing

After harvesting leaves tissues, total genomic DNA was extracted individually using the modified CTAB protocol described by Furtado [35] with slight modifications. The modifications that were made can be summarised as the following: the mixture of ground plant tissue and nuclear extraction buffer was incubated at 65 °C for 60 min with periodic mixing by inverting the tubes every 5 min; as well as the speed and time of centrifuge were increased to 4000× *g* and 7 min, respectively, after the steps of protein denaturation and DNA precipitation. However, the most vital modification in the DNA extraction procedure was the exclusion of the mixture of phenol:chloroform:isoamyl alcohol (25:24:1). The quality of DNA was assessed by NanoDrop™ 8000 Spectrophotometers (Thermo Scientific, http://www.nanodrop.com) while the DNA quantity was estimated by agarose gel electrophoresis (1%, 120 V for 1 h) based on Furtado’s study [35].

The whole genomic DNA of Iraqi rice varieties was sequenced by preparing and indexing five PCR-free libraries separately (one library for each variety), then pooling them together and sequencing over a half lane of an Illumina HiSeq 4000 flow-cell at MACROGEN (Seoul, Korea; http://dna.macrogen.com).

### 5.4. Data Downloaded for Sequence Comparisons

Raw sequence reads of 33 domesticated rice accessions were sourced from the Sequence Read Archive (SRA)-NCBI website (https://www.ncbi.nlm.nih.gov/sra) using “Download/Search for Reads in SRA” tool on CLC Genomics Workbench version 11.0.1 (CLC Bio, a QIAGEN Company, Aarhus, Denmark; www.clcbio.com). All of the species, except one, were Asian rice (*O. sativa*) relatives. *O. glaberrima*, an African rice, was included as an out-group. All related information such as the sample unique ID, project accession, species, country of origin, and ecotype was obtained from an earlier study [26], as shown in Table 2.

### 5.5. Data Processing

The raw reads of both sequenced and downloaded data were subjected to quality control (QC) analysis using the “Create Sequencing QC Report” tool in the CLC Genomics Workbench, which was used to verify the integrity of the data and determine the appropriate trimming score. The low-quality reads were trimmed at a quality limit of 0.01 and a minimum PHRED score of 25 “Trim Sequences” tool on the CLC.

### 5.6. Chloroplast Genome Assembly

A chloroplast genome of the domesticated rice was assembled and validated using a dual pipeline approach: (1) mapping assembly (MA), and (2) *de novo* assembly (*d*A) [36]. In the mapping assembly (MA) pipeline, the trimmed reads were mapped against the reference, which is *O. sativa* sub sp. japonica Nipponbare “GenBank: GU592207.1”, using “Map reads to reference” tool at three various fraction settings of length-fraction and similarity-fraction (1) 0.8 and 0.8, (2) 1 and 0.8, and (3) 1 and 0.9, this step was known as “R”. Additionally, in an attempt to mend the Cp map, two tools, “InDels and Structural Variants” and “Local Realignment”, were applied. This step was named “S”. All the analyses of mapping assembly were performed on the CLC Genomics Workbench 11.0.1.

In the *de novo* assembly pipeline, the Fast “F” model was used with combinations of Word “W” and Bubble “B” settings. Contigs generated by *de novo* were blasted against the Cp reference *O. sativa* sub sp. japonica Nipponbare “GenBank: GU592207.1” to select the Cp-exclusive contigs, and they were then updated using the “Map Reads to Contigs” tool on the CLC Genomics Workbench 11.0.1. Lastly, the updated contigs were aligned to a reference sequence to recognise overlaps and gaps using Clone Manager Professional 9.0 (www.scied.com). When non-overlapping contigs were produced, supplemental *de novo* assembly was conducted at various W-and B-settings to plug all gaps by creating additional contigs, and then all the overlapping contigs were subjected to the further analysis.

An additional improvement process was performed on both the mapping and *de novo* assembly pipelines. The improvement (Imp) process was similar to the mapping assembly (MA) pipeline, repeated twice, Imp-1 and Imp-2, with one difference, the consensus generated from each process would be a reference for the following process. The sequences of both improved Cp consensus generated by the mapping and *de novo* improvement processes were compared to identify all mismatches and then were manually corrected by reference to the reads; this step was named “manual-curation” (Appendix A). Eventually, the Cp sequence of each variety was ready for the phylogenetic analysis.

### 5.7. Phylogenetic Analysis

The consensus chloroplast sequences of the Iraqi rice and the other domesticated rice accessions were used to perform a phylogenetic analysis using Geneious software version 9.1.8 (https://www.geneious.com). The multiple alignment was conducted using the plugin MAFFT Alignment [37] with default parameters; subsequently, to analyse evolutionary relationships; the phylogenetic tree was constructed through software that roots the constructed tree based on the outgroup method: MrBayes [38], and Geneious Tree Builder. The distance between the chloroplast genomes of Iraqi and comparative rice was determined by detecting all the variants using the “variant/SNP detection” tool on Geneious software and then counting the differences (number of bases which are not identical), one of the outputs of the phylogenetic tree construction process.

### 5.8. Phylogenetic Analysis of the Nuclear Genome

An evolutionary relationship analysis at the level of the nuclear genome was undertaken using the CLC Genomics Workbench 11.0.1 and Geneious software version 9.1.8; this analysis started with the nuclear genome assembly (NGA) pipeline (Appendix A). In NGA pipeline, the “Map Reads to Reference” tool was used to map the trimmed reads of the Iraqi rice (Table 1), and the domesticated rice accessions from Asia and Africa (Table 2) against the reference, which is *O. sativa* sub-spp. Japonica cv Nipponbare “GenBank: IRGSP1.0”, applying the following setting: length-fraction of 1 and similarity-fraction of 0.8. After mapping, the consensus sequence of a whole genome for each variety was extracted using the “Extract Consensus Sequence” tool, and from that, the genome and coding sequence (CDS) tracks were generated by the “Convert to Tracks” tool. By investigating the CDS tracks for all varieties, a subset of 916 genes was identified in all varieties, and then the nucleotide sequences of 916 CDS were separately extracted from the genomes using the “Extract Annotations” tool. At the final stage of the nuclear genome assembly (NGA), all the nucleotide sequences of the 916 CDS selected from each genome were concatenated into a super-matrix of 621,012 bp by the “Join Sequences” tool. The super-matrices of all varieties were then aligned using multiple alignments MAFFT [37] on Geneious at default parameters; the alignment output was used in the following phylogenetic inference. A phylogenetic tree was constructed and rooted using the outgroup methods which are MrBayes [38], and Geneious Tree Builder (https://www.geneious.com); the default tree search settings were applied for both methods.

## Figures and Tables

**Figure 1 plants-08-00481-f001:**
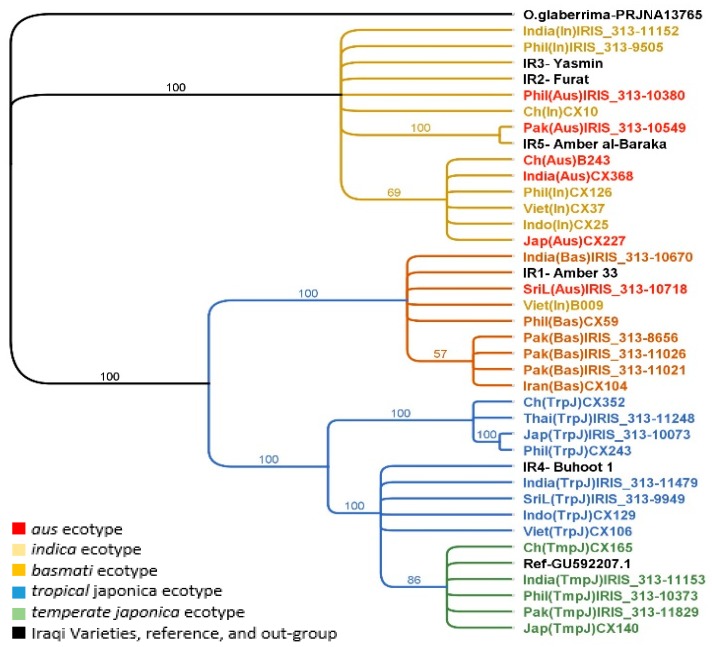
Phylogenetic relationships among chloroplast genomes of thirty-nine rice accessions. Tree topology based on MrBayes software (branch labels represent probability percentage).

**Figure 2 plants-08-00481-f002:**
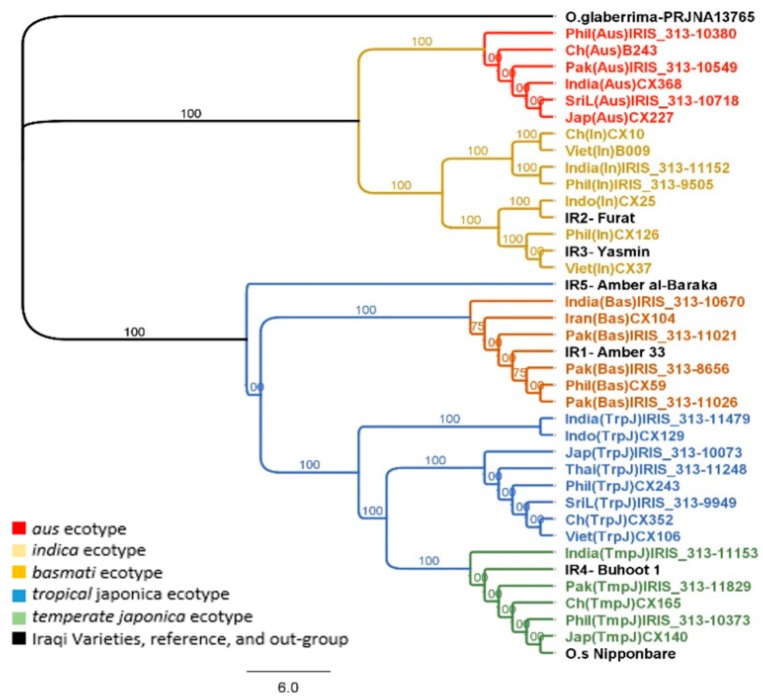
Evolutionary relationships among the multiple alignment of 916 concatenated nuclear genes of domesticated rice. Tree topology based on MrBayes software (branch labels represent probability percentage).

**Table 1 plants-08-00481-t001:** The Iraqi plant materials used in this study.

Varieties	History	Varietal Group	BioProject ID	BioSample Accessions
Amber33	Local (Iraq)	Aromatic, medium grain type	PRJNA576935	SAMN13014963
Furat	Introduced from (Vietnam) in 1996	Aromatic, medium grain type	PRJNA576935	SAMN13014964
Yasmin	Introduced from (Vietnam) in 1998	Aromatic, medium grain type	PRJNA576935	SAMN13014965
Buhooth1	Improved	Non-Aromatic, long grain type	PRJNA576935	SAMN13014966
Amber al-Baraka	Improved	Aromatic, long grain type	PRJNA576935	SAMN13014967

**Table 2 plants-08-00481-t002:** Summary of data downloaded for sequence comparisons.

No	Sample Unique ID	Project Accession	Species	Country of Origin	Ecotype *	Alignment Name (in Figure 1 and Figure 2) *
1	B243	ERP005654	*O. sativa*	China	Aus	Ch(Aus)B243
2	CX165	ERP005654	*O. sativa*	China	TmpJ	Ch(TmpJ)CX165
3	CX352	ERP005654	*O. sativa*	China	TrpJ	Ch(TrpJ)CX352
4	CX10	ERP005654	*O. sativa*	China	In	Ch(In)CX10
5	CX368	ERP005654	*O. sativa*	India	Aus	India(Aus)CX368
6	IRIS_313–10670	ERP005654	*O. sativa*	India	Bas	India(Bas)IRIS_313-10670
7	IRIS_313–11153	ERP005654	*O. sativa*	India	TmpJ	India(TmpJ)IRIS_313-11153
8	IRIS_313–11479	ERP005654	*O. sativa*	India	TrpJ	India(TrpJ)IRIS_313-11479
9	IRIS_313–11152	ERP005654	*O. sativa*	India	In	India(In)IRIS_313-11152
10	CX129	ERP005654	*O. sativa*	Indonesia	TrpJ	Indo(TrpJ)CX129
11	CX25	ERP005654	*O. sativa*	Indonesia	In	Indo(In)CX25
12	CX104	ERP005654	*O. sativa*	Iran	Bas	Iran(Bas)CX104
13	CX227	ERP005654	*O. sativa*	Japan	Aus	Jap(Aus)CX227
14	CX140	ERP005654	*O. sativa*	Japan	TmpJ	Jap(TmpJ)CX140
15	IRIS_313–10073	ERP005654	*O. sativa*	Japan	TrpJ	Jap(TrpJ)IRIS_313-10073
16	IRIS_313–10549	ERP005654	*O. sativa*	Pakistan	Aus	Pak(Aus)IRIS_313-10549
17	IRIS_313–11021	ERP005654	*O. sativa*	Pakistan	Bas	Pak(Bas)IRIS_313-11021
18	IRIS_313–11026	ERP005654	*O. sativa*	Pakistan	Bas	Pak(Bas)IRIS_313-11026
19	IRIS_313–8656	ERP005654	*O. sativa*	Pakistan	Bas	Pak(Bas)IRIS_313-8656
20	IRIS_313–11829	ERP005654	*O. sativa*	Pakistan	TmpJ	Pak(TmpJ)IRIS_313-11829
21	IRIS_313–10380	ERP005654	*O. sativa*	Philippines	Aus	Phil(Aus)IRIS_313-10380
22	CX59	ERP005654	*O. sativa*	Philippines	Bas	Phil(Bas)CX59
23	IRIS_313–10373	ERP005654	*O. sativa*	Philippines	TmpJ	Phil(TmpJ)IRIS_313-10373
24	CX243	ERP005654	*O. sativa*	Philippines	TrpJ	Phil(TrpJ)CX243
25	IRIS_313–9505	ERP005654	*O. sativa*	Philippines	In	Phil(In)IRIS_313-9505
26	CX126	ERP005654	*O. sativa*	Philippines	In	Phil(In)CX126
27	IRIS_313–10718	ERP005654	*O. sativa*	Sri Lanka	Aus	SriL(Aus)IRIS_313-10718
28	IRIS_313–9949	ERP005654	*O. sativa*	Sri Lanka	TrpJ	Sril(TrpJ)IRIS_313-9949
29	IRIS_313–11248	ERP005654	*O. sativa*	Thailand	TrpJ	Thai(TrpJ)IRIS_313-11248
30	CX106	ERP005654	*O. sativa*	Vietnam	TrpJ	Viet(TrpJ)CX106
31	B009	ERP005654	*O. sativa*	Vietnam	In	Viet(In)B009
32	CX37	ERP005654	*O. sativa*	Vietnam	In	Viet(In)CX37
33	O.glaberrima-PRJNA13765	SRP038750	*O. glaberrima*	-	-	*O. glaberrima*

32 domesticated Asian rice accessions and one domesticated African rice as an out-group downloaded from SAR-NCBI: their unique ID, species, country of origin, and ecotype was from the study of [26]; the alignment names were generated in this study. * In: *indica* subpopulation, TrpJ: *tropical japonica* subpopulation, TmpJ: *temperate japonica* subpopulation, Aus: *aus* population, Bas: *basmati* population, Ch: China, Indo: Indonesia, Jap: Japan, Pak: Pakistan, Phil: Philippines, Sril: SriLanka, Viet: Vietnam.

**Table 3 plants-08-00481-t003:** The results of the chloroplast and nuclear genome assembly.

Varieties	Chloroplast Genome		Length of Nuclear Genome (bp)
	Length of Genome (bp)	Coverage (×)	
Amber33	134,536	2909	616,371
Furat	134,500	7819	616,190
Yasmin	134,502	5495	616,301
Buhooth1	134,550	4759	616,393
Amber al-Baraka	134,493	3651	616,310
B243	134,497	2203	616,278
CX165	134,542	8818	616,377
CX352	134,553	5305	616,324
CX10	134,503	11,466	616,274
CX368	134,504	3674	616,236
IRIS_313–10670	134,535	1669	616,369
IRIS_313–11153	134,551	1639	616,360
IRIS_313–11479	134,259	2726	616,367
IRIS_313–11152	134,503	1413	616,271
CX129	134,535	6076	616 337
CX25	134,503	5830	616,210
CX104	134,532	6784	616,348
CX227	134,504	4267	616,314
CX140	134,547	5185	616,393
IRIS_313–10073	134,556	2036	616,355
IRIS_313–10549	134,495	1636	616 324
IRIS_313–11021	134,531	1978	616,383
IRIS_313–11026	134,532	1723	616,358
IRIS_313–8656	134,532	2334	616 380
IRIS_313–11829	134,539	4164	616,389
IRIS_313–10380	134,496	1857	616,331
CX59	134,536	5913	616,370
IRIS_313–10373	134,551	1464	616,363
CX243	134,556	4375	616,362
IRIS_313–9505	134,503	968	616,283
CX126	134,503	3510	616,220
IRIS_313–10718	134,531	2332	616,324
IRIS_313–9949	134,532	3041	616 385
IRIS_313–11248	134,413	974	616,336
CX106	134,529	6145	616,339
B009	134,528	839	616,231
CX37	134,503	4836	616,258
O.glaberrima- PRJNA13765	134,542	2567	616,099

The table includes the length of the chloroplast genome, the number of bases of mapped reads, and the coverage of assembled chloroplast genome for five Iraqi varieties and 32 domesticated Asian accessions and one domesticated African rice as an out-group downloaded from SAR-NCBI. This table also shows the length of the nuclear genome.

**Table 4 plants-08-00481-t004:** Summary of the number and types of variants in the chloroplast-genomes of thirty-nine domesticated rice-accessions.

	Variation Type	SNPs	MNPs	Del	Ins	Total
Group						
Indica		37	5	10	9	61
Japonica		1	0	1	1	3
Basmati		4	1	0	1	6
Indica & Basmati		3	2	3	0	8
Indica and Basmati and Japonica		3	0	1	3	7
Total		48	8	15	14	85

## Data Availability

All NGS sequence data as raw data was submitted to NCBI at the Sequence Read Archive (SRA) and is available as SRA Submission# SUB6410326 (under BioProject# PRJNA576935 and BioSample# SAMN13014963, SAMN13014964, SAMN13014965, SAMN13014966, and SAMN13014967 represent Amber33, Furat, Yasmin, Buhooth1, and Amber al-Baraka, respectively).

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
