# Peer review of "Relationships between Iraqi Rice Varieties at the Nuclear and Plastid Genome Levels"

_plants, 2019, doi:10.3390/plants8110481_

Round 1
Reviewer 1 Report
The manuscript titled “Relationships between Iraqi Rice Varieties at the Nuclear and Plastid Genome Levels” with the Manuscript ID plants-606722 the MDPI journal- Plants (ISSN 2223-7747) describes a phylogenetic study that investigates the origins of five Iraqi rice varieties, with an emphasis on variety Amber-33, which is reported to be particularly preferred for it aromatic properties in Iraq.
The study is technically sound and the conclusions are supported by the results and the previous literature. The methods utilized are robust and previously well described in the literature.
Investigations into the origins of Iraqi rice varieties are novel- and the authors explain and interpret their findings well based on the available historical information.
There are some places where some minor editorial revisions may be warranted- which I will list below. Once the issues itemized below are addressed, I have no reservations to recommend this manuscript for publication.
General:
· There are many places in the manuscript where the varieties in the study are identified by both their study identifiers (Ir1, etc.) as well as their cultivar names(Amber33, etc.). I suggest the authors pick one such identifier and use it consistently through out the manuscript. The long lists of cultivar identifiers in paragraphs distracts from the flow of the paper.
· Similarly- for the varieties whose data was obtained from NCBI-SRA, I would recommend assigning and using study identifiers that can be summarized in the Table 3 to simplify the flow while these varieties are referenced in the text( Lines 171-189).
· The discussion of specific chloroplast genome sub-regions are also intermittently referenced by their long- names (large single copy, etc.) and their acronyms (LSC, etc.) in the text. I would recommend using the acronyms after the first open-form referencing of these longer names for efficiency and flow.
Itemized :
1. Lines 52-60 present a discussion and background on the history of use of DNA marker technologies for evolutionary/ phylogenetic research. This discussion can be omitted without any loss of value to the topic being discussed, since the authors are leveraging NGS technology, as described in lines 61-64. It would suffice the historical use of these older approaches with a single sentence and references.
2. Line 126 – table caption has a typo : verities à varieties
3. Table 5 can be moved to supplementary material without loss of value- since no discussion on molecular influences of the changes (synonymous- non-synonymous, non-coding) are provided.
4. Line 274 - use “investigation” instead of “searches”
5. Line 282 – The statement on historical evolution and hybridization should be reconsidered – since rice is a naturally selfing species with very low natural hybridization rates. The observation described indicates a role for intentional breeding-introgression or crossing, as opposed to natural evolution or hybridization.
6. Line 291 – the refence to “tree of evolution” can better be referenced as the “observed phylogenetic tree”, since potentially there may be many possible configurations and tree topologies that can be suggested by same or similar genetic marker data.
7. Line 294 – statement “country of origin of rice” should be rectified as “country of origin of this variety or Amber33.
8. Lines 295-306 provides historical and etymological information on the origins of the word “Basmati” with a brief justification on why it should be classified with aromatic sub-population of indica ecotypes versus tropical japonica ecotypes. This section can be moved to the conclusion section since it does not really add to the discussion (of results).
9. Line 335 – redundant use of “were”. Please re-examine the sentence.
10. Line 336- grounded is used improperly- the verb is to grind, past tense is ground
11. Line 411- matrixes is improper – should be matrices.
Author Response
General:
· There are many places in the manuscript where the varieties in the study are identified by both their study identifiers (Ir1, etc.) as well as their cultivar names(Amber33, etc.). I suggest the authors pick one such identifier and use it consistently through out the manuscript. The long lists of cultivar identifiers in paragraphs distracts from the flow of the paper.
All (Ir1, etc.) have been changed to (Amber33, etc.)
· Similarly- for the varieties whose data was obtained from NCBI-SRA, I would recommend assigning and using study identifiers that can be summarized in the Table 3 to simplify the flow while these varieties are referenced in the text( Lines 171-189).
The study identifiers are found in Table 3 for all varieties with data that was obtained from NCBI-SRA while these varieties are referenced in the text.
· The discussion of specific chloroplast genome sub-regions are also intermittently referenced by their long- names (large single copy, etc.) and their acronyms (LSC, etc.) in the text. I would recommend using the acronyms after the first open-form referencing of these longer names for efficiency and flow.
The acronyms of (large single copy, etc.) which are (LSC, etc.) have been used after the first open-form referencing of these longer names.
Itemized :
1. Lines 52-60 present a discussion and background on the history of use of DNA marker technologies for evolutionary/ phylogenetic research. This discussion can be omitted without any loss of value to the topic being discussed, since the authors are leveraging NGS technology, as described in lines 61-64. It would suffice the historical use of these older approaches with a single sentence and references. Done
2. Line 126 – table caption has a typo: verities à varieties. Done
3. Table 5 can be moved to supplementary material without loss of value- since no discussion on molecular influences of the changes (synonymous- non-synonymous, non-coding) are provided. Done
4. Line 274 - use “investigation” instead of “searches” Done
5. Line 282 – The statement on historical evolution and hybridization should be reconsidered – since rice is a naturally selfing species with very low natural hybridization rates. The observation described indicates a role for intentional breeding-introgression or crossing, as opposed to natural evolution or hybridization. Done
6. Line 291 – the refence to “tree of evolution” can better be referenced as the “observed phylogenetic tree”, since potentially there may be many possible configurations and tree topologies that can be suggested by same or similar genetic marker data. Done
7. Line 294 – statement “country of origin of rice” should be rectified as “country of origin of this variety or Amber33. Changed
8. Lines 295-306 provides historical and etymological information on the origins of the word “Basmati” with a brief justification on why it should be classified with aromatic sub-population of indica ecotypes versus tropical japonica ecotypes. This section can be moved to the conclusion section since it does not really add to the discussion (of results). Please see “Table of changes”
9. Line 335 – redundant use of “were”. Please re-examine the sentence. Done
10. Line 336- grounded is used improperly- the verb is to grind, past tense is ground. Done
11. Line 411- matrixes is improper – should be matrices. Done
Reviewer 2 Report
The authors use chloroplast and nuclear phylogenies to investigate the evolutionary relationships of five Iraqi and 33 Asian rice varieties with a focus on the Amber33 variety. The introduction explains why Amber is of interest but the paragraphs on evolution and plant genomes are excessively basic. From the results, all the tables (perhaps with the exception of Table 4) seem like supplementary material. The nuclear and chloroplast phylogenies should be directly compared in a figure to make clear where the discrepancies are. The mapping based assembly should be the better of the two methodologies since the rice reference genome is such high quality. I don't think the de novo assemblies are of value. Finally, there are numerous errors in grammar and inconsistencies in labeling, as well as varying fonts. As the authors conclude, I think a biogeographical analysis would greatly improve this study, as well as more direct comparisons between the chloroplast and nuclear phylogenies.
Author Response
|
No. |
Reviewer 2 comments |
Original |
Changes |
Page |
|
1 |
From the results, all the tables (perhaps with the exception of Table 4) seem like supplementary material. |
Table 1, 2, 3, 4 and 5 |
Keep Table 1, 2, 3 and 4, and changed Table 5 to Table S4 (base on the recommendation of the first reviewer). We have not changed table 1, 2 and 3 as supplementary materials because table 1 & 2 are the main part of the materials and methods section while Table 3 is part of the result section. |
|
|
2 |
The nuclear and chloroplast phylogenies should be directly compared in a figure to make clear where the discrepancies are. |
Figure 1 Figure 2 |
Keep figure 1 and 2 separate because the comparison between the chloroplast and nuclear phylogenies is not one of the main objectives of this study. |
|
|
3 |
The mapping based assembly should be the better of the two methodologies since the rice reference genome is such high quality. I don't think the de novo assemblies are of value. |
|
I totally agree with this comment. However, de novo was not the only method used in this assembly, but a dual pipeline approach was applied to improve assembly quality and obtain the most accurate chloroplast genome as indicated by (Moner et al., 2018). |
|
|
4 |
Finally, there are numerous errors in grammar and inconsistencies in labeling, as well as varying fonts. |
|
Several grammar, labeling and other mistakes were corrected (please see the table of changes for the first reviewer). |
|
